# Object-Oriented Crop Classification Using Time Series Sentinel Images from Google Earth Engine

**Hanyu Xue** [1,2], **Xingang Xu** [1,2,]*, **Qingzhen Zhu** [2], **Guijun Yang** [1], **Huiling Long** [1], **Heli Li** [1], **Xiaodong Yang** [1], **Jianmin Zhang** [3], **Yongan Yang** [3], **Sizhe Xu** [1,2], **Min Yang** [1] and **Yafeng Li** [1,2]

1 Key Laboratory of Quantitative Remote Sensing in Agriculture of Ministry of Agriculture and Rural Affairs, Information Technology Research Center, Beijing Academy of Agriculture and Forestry Sciences, Beijing 100097, China
2 School of Agricultural Engineering, Jiangsu University, Zhenjiang 212013, China
3 Tianjin Development and Demonstration Center for High-Quality Agricultural Products, Tianjin 301508, China
* Correspondence: xuxg@nercita.org.cn

**Abstract:** The resulting maps of land use classification obtained by pixel-based methods often have salt-and-pepper noise, which usually shows a certain degree of cluttered distribution of classification image elements within the region. This paper carries out a study on crop classification and identification based on time series Sentinel images and object-oriented methods and takes the crop recognition and classification of the National Modern Agricultural Industrial Park in Jalaid Banner, Inner Mongolia, as the research object. It uses the Google Earth Engine (GEE) cloud platform to extract time series Sentinel satellite radar and optical remote sensing images combined with simple noniterative clustering (SNIC) multiscale segmentation with random forest (RF) and support vector machine (SVM) classification algorithms to classify and identify major regional crops based on radar and spectral features. Compared with the pixel-based method, the combination of SNIC multiscale segmentation and random forest classification based on time series radar and optical remote sensing images can effectively reduce the salt-and-pepper phenomenon in classification and improve crop classification accuracy with the highest accuracy of 98.66 and a kappa coefficient of 0.9823. This study provides a reference for large-scale crop identification and classification work.

**Keywords:** Sentinel images; object-oriented; SNIC algorithm; random forest classification; support vector machine classification; Google Earth Engine

## 1. Introduction

The timely and effective acquisition of regional crop cultivation distribution information is of great significance to a country in formulating food policies and guaranteeing national food security, etc. Satellite remote sensing technology has the characteristics of a wide monitoring range, strong periodicity, and diverse spectral information. With the development of satellite remote sensing technology and the improvement of the temporal and spatial resolutions of remote sensing images, carrying out large-scale crop classification identification, area extraction, and planting structure change analysis based on remote sensing image resolution technology has become an important element of agricultural remote sensing yield estimation applications.

Currently, crop classification and identification methods using time series images using image-oriented are widely applied [1–5]. HJ-CCD time series optical remote sensing images were used to construct a decision-tree-based classification model with the spectral vegetation index time series variation characteristics of crops for the effective classification and identification of multiple crop plantings [1]. Multitemporal RADARSAT-2 fully polarized SAR time series images were used to achieve the efficient extraction of the phenological period of rice based on the time series curve variation characteristics of the polarization

feature parameters of rice [2]. Phan Thanh Noi et al. [6] used Sentinel-2 image data to compare the performance of classifiers, such as RF, kNN, and SVM, for land use cover classification. The above multitemporal and multifeature classification methods based on pixels often carry out crop classification and recognition by extracting the temporal optical (or microwave) features of image elements. They used several methods and tested them to determine the best. Although they achieve high classification accuracy, to a certain extent, they usually ignore the spatial correlation between adjacent image elements [7], which is prone to salt-and-pepper noise [8]. Salt-and-pepper noise exists in most pixel-based classifications with high-resolution images. The essence of this phenomenon is the misclassification of pixels affected for various reasons. The object-oriented classification method based on remote sensing images can reduce the salt-and-pepper noise to a certain extent [9,10]. D. Geneletti et al. [11] used a maximum likelihood classifier and additional empirical rules to classify TM images and then segmented the orthoimages. Then, they used the previously classified TM images as a reference to perform the classification of the segmented images. It has been shown that the object-oriented approach has some improvements in classification accuracy compared to the image element-based classification methods [12,13]. In recent years, Google Earth Engine (GEE), as an open cloud platform with powerful data processing, analysis, storage, and visualization capabilities, has been widely used in the field of remote sensing research and plays an active role in research on remote sensing identification and extraction methods for crop classification. Some scientific researchers in the field of remote sensing have conducted many classification studies based on the GEE cloud platform. K. Zhou et al. [14] extracted several winter wheat NDVI features based on the GEE cloud platform and calculated the winter wheat planting area using the random forest classification method. T. N. Phan et al. [15] used Landsat 8 surface reflectance (L8sr) data with eight different combination strategies to produce and evaluate land cover maps for a study area in Mongolia, implementing the experiment on the GEE platform with a widely applied algorithm, the random forest (RF) classifier, to analyze the effect of different composition methods, as well as different input images, on the resulting maps. H. Zhang et al. [16] used multitemporal environmental star HJ-1A/BCCD data and its multiperiod smoothed and reconstructed NDVI time series curve features to classify crops at the object scale using a decision tree algorithm. B. Du et al. [17] used Sentinel-2A NDVI time series features and combined object-oriented classification and support vector machine (SVM) classification based on the Google Earth Engine (GEE) platform. C. Luo et al. [18] segmented the composite images with simple noniterative clustering (SNIC) according to different sizes and input the training samples and processed images into a random forest classifier for crop classification.

The above studies show that current studies have used object-oriented methods to improve crop classification accuracy to some extent, but most of them tend to use a single type of remote sensing image, such as time series visible–near–infrared optical images or microwave radar image features, with less consideration of the complementary advantages between different types of images.

Google Earth Engine (GEE) is a tool developed by Google that stores publicly available remote sensing image data based on its millions of servers around the world and state-of-the-art cloud computing and storage capability, enabling GEE users to easily extract, call, and analyze massive remote sensing big data resources, providing huge potential for large-scale and long-term remote sensing analysis [19,20].

In this paper, based on Google Earth Engine cloud computing technology, an SNIC object-oriented multiscale segmentation algorithm was used to carry out large-scale object-oriented crop classification application research using classification methods such as random forest (RF) and support vector machine (SVM) and integrating the feature of Sentinel-1 SAR microwave radar and Sentinel-2 optical remote sensing images.

## 2. Materials and Methods

### 2.1. Study Area and Data Source

2.1.1. Study Area

The study area is located in the eastern part of Jalaid Banner, Hinggan League, Inner Mongolia Autonomous Region, at the junction of Heilongjiang, Jilin, and Inner Mongolia provinces. The terrain is gentle and mostly plain, with elevations ranging from 150 to 250 m. The four seasons are distinct, with a temperate continental climate and concentrated precipitation, with the greatest amount of precipitation in summer, accounting for more than 70% of the average annual precipitation. The study area has a well-developed water system, with the second largest river in Inner Mongolia, the Chol River, in the north, numerous streams and artificial water channels in the study area, and abundant groundwater resources. The study area is very suitable for agricultural activities because of its abundant light and water resources. Figure 1 shows the location and topography of the study area, as well as sample distribution.

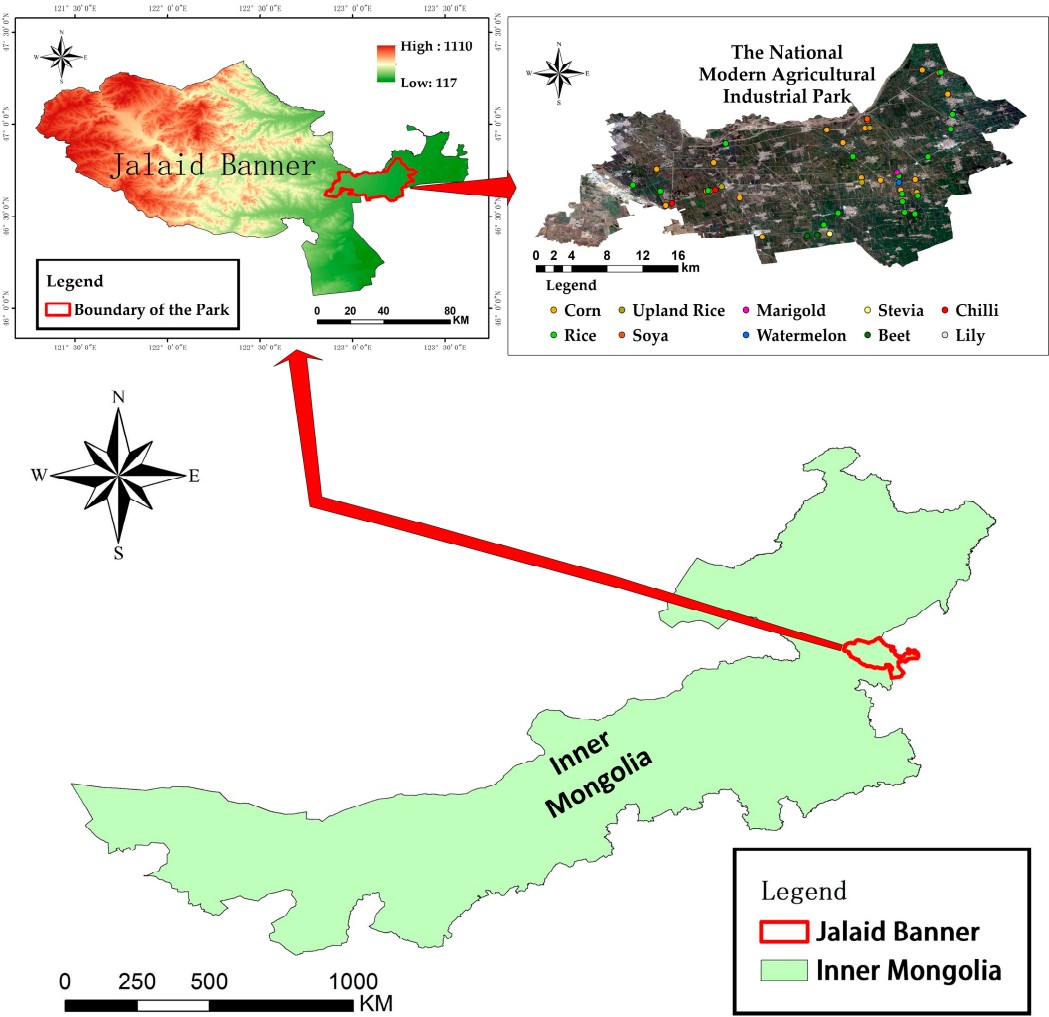

**Figure 1.** Topographic map of the Modern Agricultural Industrial Park in Jalaid Banner. The geographical location and topographic information of Jalaid Banner are displayed, as well as the location of the park and the distribution of the sample points.

The study area mainly grows four kinds of crops, which are rice, upland rice, soya, and corn. Table 1 lists the growth period information of the four major crops in the study area. The main growth period of these crops is from early May to early September.

**Table 1.** The growth period of major crops in the study area. E means the early 10 days of a month, M is the middle 10 days, and L represents the later 10 days.

| Name | May | | | June | | | July | | | August | | | September | | | October |
|---|---|---|---|---|---|---|---|---|---|---|---|---|---|---|---|---|
| | E | M | L | E | M | L | E | M | L | E | M | L | E | M | L | E |
| Rice | | | Sowing | | Tillering | | | Heading | | | | Filling | | Maturity | | |
| Upland Rice | | Sowing | | | Tillering | | | | | Heading | | Filling | | | | Maturity |
| Soya | Sowing | | | | Seeding | | | Flowering | | Podding | | | Filling | | | Maturity |
| Corn | Sowing | | Seeding | | | | Jointing | | | Tasseling | | Filling | | | | Maturity |

### 2.1.2. Image Dataset and Preprocessing

This study used Sentinel-2 spectral remote sensing data and Sentinel-1 SAR remote sensing data to extract different crop-growing areas. Their different bands have different resolutions up to 10 m. The revisit period of a single satellite is 10 days, and the two satellites complement each other, so the revisit period is 5 days. Given that its spectral, spatial, and temporal resolutions meet the basic requirements of this classification study, Sentinel-2 images were first selected as the basic spectral image data for the classification. The study area mainly grows corn, rice, and upland rice crops, so the remote sensing image data should also be selected within the fertility period of these major crops. At the same time, special attention should be paid to the quality of the images themselves. From early May to late July, the study area is in the rainy season, with high cloudiness and high coverage of the ground spectral information. based on the image quality, three-view spectral remote sensing images were selected for this study on 21 May, 10 June, and 25 June 2021.

In June and July, the cloudiness in Jalaid Banner, Hinggan League, Inner Mongolia, was high, and its coverage of the ground spectral information was large. The Google Earth Engine (GEE) cloud platform integrates zenith reflectance (TOA) and atmospherically corrected surface reflectance (SR) products, which do not require geometric and atmospheric corrections again. This is one of the advantages of the GEE cloud platform. Subsequently, the part covered by the cloud must be removed.

The Sentinel-1 SAR satellite, consisting of two satellites, A and B, with a minimum revisit period of 6 days, is based on C-band (microwave) remote sensing imaging with four imaging modes (stripmap (SM), interferometric wide swath (IW), extra wide swath (EW), and wave (WV)). The satellite image width is 400 km, which has the advantages of multipolarization and a short revisit period. Meanwhile, based on the characteristics of microwave remote sensing itself, it has a strong penetration of clouds, so it is especially suitable for the requirements of cloudy and rainy climates in this study area. The multiple Sentinel-2 spectral remote sensing images with large clouds and relatively poor quality during the crop fertility period in the National Modern Agricultural Industrial Park in Jalaid Banner, Inner Mongolia, cannot be utilized for input into the classifier. The inclusion of Sentinel-1 SAR remote sensing images is making up for the deficiency of Sentinel-2 spectral remote sensing images, and this will also make the input image data more diversified, thus achieving the purpose of improving the classification accuracy. Microwave remote sensing images from 5 June to 2 September 2021 were used for the study. Table 2 shows all of the images used in the study.

Sentinel-1 SAR microwave remote sensing data has strong penetration of the atmosphere, and it is sensitive to the texture features of the ground [21]. However, due to the fact of its imaging method and characteristics, the topography of the target study area will interfere with the imaging, and when there are higher ground objects, such as trees and houses, some shadows will be generated next to these features, which will interfere with the analysis of remote sensing images. At the same time, the imaging process is also affected by noise, the source of which may be both the surrounding features, the atmosphere, and the remote sensor itself. Therefore, the preprocessing of Sentinel-1 SAR images needs to address the above issues. Digital elevation models (DEMs) can represent the elevation changes of regional topography, and a large number of topographic factors

can also be extracted based on such data to provide an analysis and reference basis for the study. This study utilized SRTM DEM 30 m data for terrain correction and shadow removal of Sentinel-1 SAR data in the study area. The study calculated the shadow areas and removed them based on the local incidence angle, solar altitude angle, and slope of the target image element location. Then, the multiview images of the same area in a short period were averaged to reduce the errors caused by noise. Finally, this data source was used to calculate the polarized VV and VH values and form a complete image.

**Table 2.** Sentinel images used in this study.

| Num | Sensor | Date | Quantity |
| --- | --- | --- | --- |
| 1 | S1GRD | 5 June 2021 | 1 |
| 2 | S1GRD | 12 June 2021 | 2 |
| 3 | S1GRD | 17 June 2021 | 1 |
| 4 | S1GRD | 24 June 2021 | 2 |
| 5 | S1GRD | 29 June 2021 | 1 |
| 6 | S1GRD | 6 July 2021 | 2 |
| 7 | S1GRD | 11 July 2021 | 1 |
| 8 | S1GRD | 18 July 2021 | 2 |
| 9 | S1GRD | 23 July 2021 | 1 |
| 10 | S1GRD | 3 July 2021 | 2 |
| 11 | S1GRD | 4 August 2021 | 1 |
| 12 | S1GRD | 11 August 2021 | 2 |
| 13 | S1GRD | 16 August 2021 | 1 |
| 14 | S1GRD | 23 August 2021 | 2 |
| 15 | S1GRD | 28 August 2021 | 1 |
| 16 | S1GRD | 2 September 2021 | 2 |
| 17 | S2 | 21 May 2021 | 1 |
| 18 | S2 | 10 June 2021 | 1 |
| 19 | S2 | 25 June 2021 | 1 |

2.1.3. Field Survey Sample

The samples for this study were field survey data, and the sample points covered the entire study area. The person in charge of the park acted as a guide to ensure a complete sample of major crops and a rich sample of other crops, as well as to ensure a uniform spatial distribution of the major crops.

In addition, the discrimination of crop types based on spectral features is one of the main ways of crop remote sensing classification identification and extraction research at present. Using Sentinel-2 spectral features of different crops, crops can be effectively distinguished from other features. Therefore, this study used the standard false-color composite method and the method of combining different bands to form false-color composite images to assist in verifying the accuracy of the crop samples.

The spectral identification of corn, one of the main crops grown in the study area, was used as an example. As shown in Figure 2, the study distinguished vegetation from other features by standard false color composite (B8, B4, and B3). The B8, B11, and B12 bands of Sentinel-2 were also used. The B8 band is the near-infrared band with a central wavelength of 0.842 μm; B11 and B2 bands are short-wave infrared bands with central wavelengths of 1.610 and 2.190 μm, respectively. B8, B11, and B12 were put into the RGB band to obtain the B8, B11, and B12 false color composites. Rice is also grown in a considerable area in the study area. The study used B11, B8, and B4 (red band) for the false color composite of images in the study area.

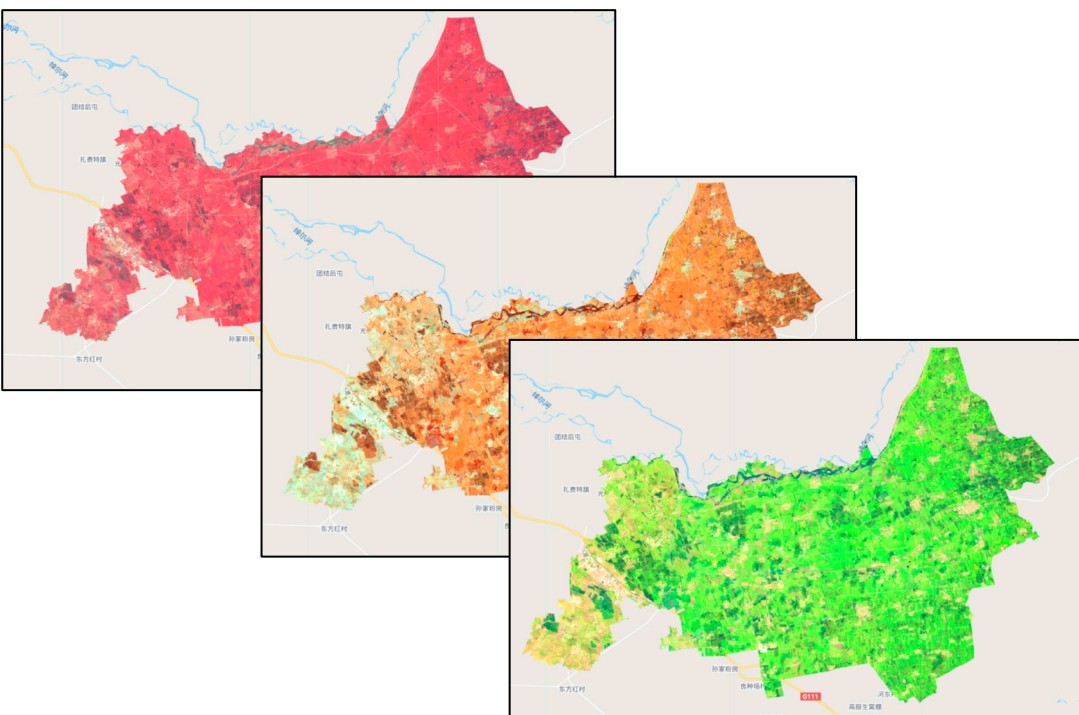

**Figure 2.** False color composite images.

The method of false color composites using different bands of Sentinel-2 makes the sample testing easier, and adding, deleting, and adjusting the samples before they are entered improves the accuracy of the samples and the classification. It also provides realistic and feasible technical support for adding samples in the absence of field survey samples.

### 2.1.4. Other Materials

The planting of major crops in the study area is characterized by a large area, concentrated distribution, and obvious boundaries. Therefore, according to the characteristics of the study area and the actual needs, the study used crop plot boundary data within the modern agricultural industrial park of Jalaid Banner instead of the commonly used complete study area boundary data. The advantage of this approach is that it eliminates a variety of irrelevant land use types in the study area, reduces the complexity of classification due to the input of other irrelevant features such as residential houses, roads, trees, and water systems, and then the crop types in the study area are classified with higher accuracy.

### *2.2. Technical Process*

### 2.2.1. Research Process

Based on Sentinel-2 Sentinel-1 SAR image data from the GEE cloud platform, the study uses the SNIC image segmentation algorithm to segment the images. Then, the crops in the study area were extracted using the random forest method or support vector machine method for classification. The steps are as follows: (1) obtain Sentinel-2 and Sentinel-1 SAR datasets from the platform and perform preprocessing operations, such as screening and declouding; (2) construct vegetation index time series curves based on spectral features and VV and VH time series curves based on backscattering and extract the most optimal feature set by combining the pseudo-color composite map of the study area; (3) use the SNIC image segmentation algorithm to segment images and form an image dataset; (4) combine the image dataset with the sample dataset as input and train the classifier; (5) use the random forest method with support vector machine to classify; (6) accuracy verification. The specific experimental process is illustrated in Figure 3.

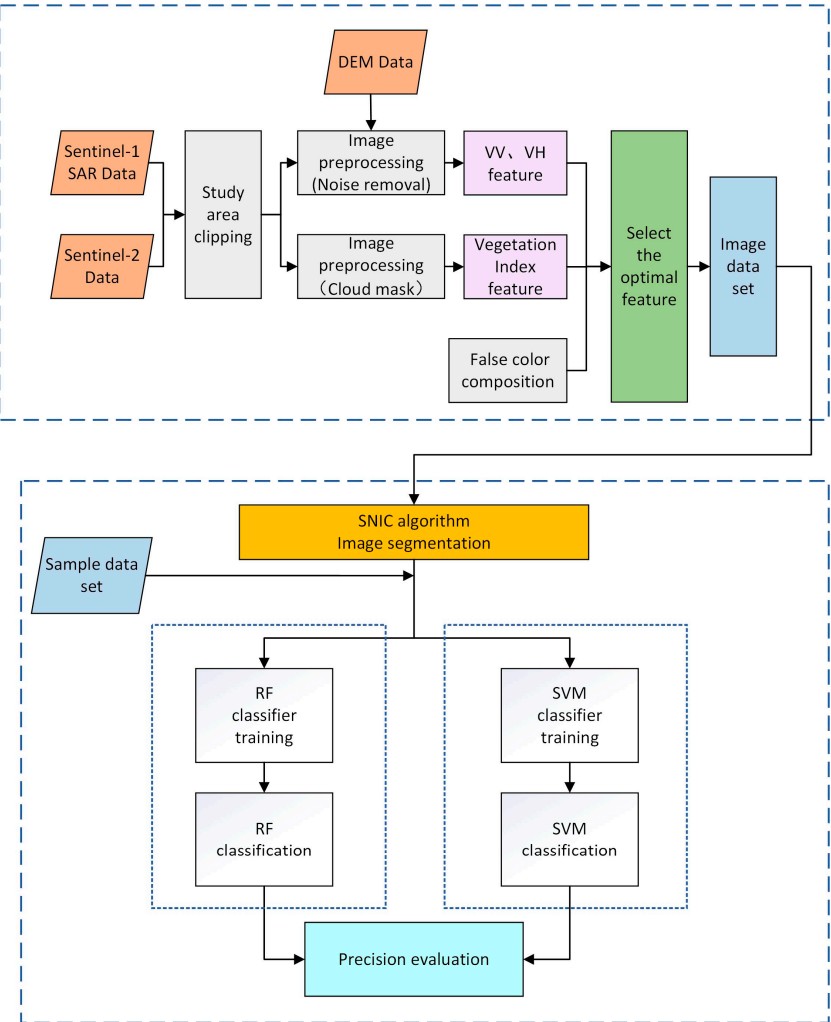

**Figure 3.** Research process.

### 2.2.2. SNIC Image Segmentation

The traditional pixel-based classification method suffers from misclassification, omission, and salt-and-pepper noise; especially for Sentinel-1 SAR remote sensing data, the salt-and-pepper noise is very serious. Usually, to ensure the final classification effect, a denoising step is needed again for the classified images. Object-oriented algorithms solve this problem at the source by considering the neighborhood information of a given pixel and dividing the image into some fixed "regions" or "targets" according to certain parameters or forming many superpixels [22–24].

The concept of superpixel is an image segmentation technique proposed and developed by Xiaofeng Ren in 2003 [25], which refers to an irregular block of pixels with certain visual significance composed of neighboring pixels with a similar texture, color, luminance, and other characteristics.

Rather than continuing the microscopic subdivision based on ordinary pixels, superpixels are a collection of pixels. These pixels have similar color, texture, and other features, and they aggregate similarly located pixels with similar features into a locally uniformly connected region.

Superpixel segmentation can simplify an image from millions of pixels to approximately two orders of magnitude fewer superpixels, which can effectively reduce redundant image information and speed up subsequent change detection processing, and because the local spatial neighborhood information of the image is considered, superpixels have

a certain noise suppression capability [26]. One of the more commonly used methods is SLIC, with its improved version—SNIC.

The SNIC algorithm [27] is a version of the SLIC algorithm [28] after being improved by ACHANTA et al. in 2017. SNIC initializes the clustering centers in the same way as SLIC. The affinity of a pixel to a clustering center is measured as a distance in a five-dimensional space of color and spatial coordinates. The SNIC algorithm uses the same distance metric as SLIC. This distance combines the normalized spatial distance and the color distance. When the spatial location $x = [x, y]^T$ and CIELAB color $c = [l, a, b]^T$, the distance from the $k$-th superpixel clustering center $C[k]$ to the $i$-th candidate pixel is:

$$d_{i,k} = \sqrt{\frac{x_i - x_{k2}^2}{s} + \frac{c_i - c_{k2}^2}{m}} \tag{1}$$

For an image with $N$ pixels, each $K$ superpixel is expected to contain $N/K$ pixels. Assuming that the superpixels are square, the value of superpixel edge length s is set to $\sqrt{N/K}$. Another parameter value, $m$, is called the compactness factor and is provided by the user. A higher value of $m$ leads to a more compact and regular superpixel at the cost of poorer boundary attachment and vice versa.

The SNIC algorithm uses a priority queue, Q, to select the next pixel to be added to the superpixel to ensure that strong connectivity is achieved at the beginning of the algorithm. The algorithm is as follows:

First, initialize $K$ seeds $C[K] = \{x_k, c_k\}$ and use these seeds to create $K$ elements $e_i = \{x_i, c_i, k_i, d_{i,k}\}$, From the above, $i$ represents the element node, $x_i$ represents the spatial location of this pixel point, $c_i$ represents the color information of this pixel, $k$ represents the label ordinal number of each superpixel from 1 to $k$, $d_{i,k}$ as in the above equation represents the distance from the $k$-th superpixel clustering center $C[k]$ to the $i$-th candidate pixel, and the initial value is set to 0. Arrange these elements according to the distance $d_{i,k}$ between element $e_i$ and the $k$-th superpixel, and the smaller the distance, the higher the priority.

When queue Q is nonempty, the element with the highest priority is popped. When the element is not tagged at the corresponding position in the tagging graph, it is tagged with a superpixel tag. In addition, calculate its 4 or 8 neighborhood pixels that have not been tagged yet, create a new element with the tag set to k, calculate the distance to the center of that tagged superpixel, and put it into queue Q by priority. Iterate over it until queue Q is empty.

The settings of the parameters need to be considered when using the SNIC algorithm in GEE. "Size" is the pixel-based spacing of the superpixel seed positions, i.e., the segmentation size, which is set according to the study area. "Compactness" indicates the degree of compactness. The higher the value, the closer the segmentation result is to a square. However, since most of the plots in the study area are rectangular, the "compactness" is set to 0. "Connectivity" stands for connectivity, which is set to 8 in this study. The "seed" refers to the number of image elements that are the center of the segmentation [18]. Since the spacing has been set before, the "seed" parameter is not set again. The biggest influence on the classification effect is the "size" parameter. It should be noted that the larger or smaller this value, the better the classification effect. The image collection of all Sentinel images was put in. According to the correlation between the pixels, the images were segmented by the SNIC algorithm.

### 2.2.3. Random Forest Classification

In machine learning, a random forest [29] is a classifier that contains multiple decision trees. As an excellent machine learning method, it has been successfully applied to many fields. Not only can random forests solve classification and regression problems, but they have also attracted increasing attention in the field of feature selection. In the 1980s, Breiman et al. invented the decision tree algorithm [30], which can reduce the compu-

tational effort by repeatedly dichotomizing data for classification or regression. In 2001, Breiman et al. combined the decision trees to form a random forest [31].

The training sample set is the basis for training the classifier, and the quality of the sample set directly affects that of the resulting maps. To establish a high-quality training set, there are generally three requirements. First, the classification category is uniform; second, all kinds of samples have high representativeness; and, third, all kinds of samples should be evenly distributed in the study area. Based on the above requirements, the sample training set is built first, and it is necessary to satisfy that each class of samples is distributed in the study area and as uniformly as possible while also ensuring sample diversity and completeness. Next, the classifier can be trained by entering all samples from the root node into a tree; a splitting criterion is included at the root node and each intermediate node, splitting the samples in that node into two child nodes. The training process of the decision tree is based on attribute value testing splitting the input training set into subsets and repeating the splitting in a recursive manner in each split into subsets until it stops when all elements in the subset at a node have the same value or when the attribute values are exhausted or other given stopping conditions.

The GEE cloud platform integrates the random forest algorithm, which can be called directly in the process of remote sensing information extraction. In addition, the number of feature variables, m, and the number of decision trees, n, are the main factors that affect the classification accuracy of the random forest algorithm under the condition of certain training samples. For the random forest method, there were 88 features in the feature collection and 500 trees in the forest. Using the GEE cloud platform for random forest classification is easy to operate and fast to process.

### 2.2.4. Support Vector Machine Classification

Support vector machine (SVM) [32] is a machine learning method based on the statistical theory that was developed in the mid-1990s. It was first proposed by Vapnik in 1995 [33]. It finds a hyperplane (or hypersurface) of the data in a high-dimensional space by some sum function that maximizes the distance from the nearest point to this surface. The support vector machine approach to classification then builds on this by partitioning the dataset to be classified into multiple discrete categories that are consistent with the form of the training samples, maximizing the distance between the categories.

SVM was originally proposed to solve binary classification problems, but when dealing with multiclass problems, it is necessary to build a suitable multiclassifier. Currently, there are two methods: "one-versus-rest" and "one-versus-one". The one-versus-rest method (OVR SVMs) classifies the samples of one class into one class and the remaining samples into another class so that k SVMs are constructed for k classes of samples, and the unknown samples are classified into the class with the largest classification function value. The one-versus-one method (OVO SVMs, or pairwise) is used to design an SVM between any two classes of samples so that $n(n - 1)/2$ SVMs are designed for n classes of samples. The result is that the one with the most votes is the result of this classification. The libsvm method used in the GEE is the one-to-one method.

The commonly used kernel functions in SVM include linear kernel function (linear), polynomial function (poly), sigmoid function (sigmoid), and radial basis function (RBF). Among them, RBF is widely used for its good performance in processing nonlinear data. Therefore, the libsvm method of the RBF kernel was used for the classification experiments.

In this study, the overall accuracy of the classification obtained after the image dataset was input into the SVM classifier was below 90%. To improve the classification accuracy and increase the classification efficiency, the input image data need to be normalized first before the SVM method is classified [34–38]. The results show that the classification accuracy of the input classifier training after normalizing the image data has been greatly improved. Meanwhile, because of the high feature dimensionality of the image data, this study conducted a principal component analysis on the input normalized image dataset. The feature dimensionality was reduced to improve the classification speed. For the

support vector machine method, the libsvm method with the RBF kernel was used for the classification experiments. The "gamma" was set to 0.1, and "cost" was set to 100 in the RBF kernel.

### 2.2.5. Validation Strategy

The cross-validation was used in the study. Some samples were selected in the training set to test the model. A part of the training set data was retained as the test set, and the parameters generated by the training set were tested to determine the degree of conformity of these parameters to the data outside the training set relatively objectively. The whole dataset was fixed into a training set and a test set. The samples were randomly selected in the proportion of 7:3 in this study. That is, 70% of all samples were used for the training set and 30% for the test set. After all classifications were completed, a confusion matrix was used to analyze the accuracy, and the kappa coefficient was used to test the consistency. Finally, producer accuracy, user accuracy, and overall accuracy were calculated in this study.

$$Kappa = \frac{p_o - p_e}{1 - p_e} \tag{2}$$

$$p_o = \frac{\sum_{i=1}^{n} a_{nn}}{N} \tag{3}$$

$$p_e = \frac{\sum_{i=1}^{n} a_{i+} * a_{+i}}{N^2} \tag{4}$$

## 3. Image Feature Analysis and Study Results

### 3.1. Feature Selection Based on Sentinel-2 Spectral Data

Different crops possess different spectral characteristics. The spectral data were used to distinguish different crops by their specific spectral features. In this chapter, the Sentinel-2 spectral data were used to classify the crops in the National Modern Agricultural Industrial Park in Jalaid Banner, Inner Mongolia, according to their growth cycles and phenological stages.

For the classification work using remote sensing images, it is difficult to accurately classify the features and crops in the study area by only using NDVI data of one view image as the input data. If only NDVI data from one image is used for classification, it will lead to serious misclassification and omission and, eventually, lead to very low classification accuracy. According to the research trend in recent years, it is a good choice to use NDVI time series data. However, after initial attempts, it was found that the classification results obtained by using NDVI time series data of the whole crop reproductive period in the study area as the input data were still unsatisfactory. After repeated checks and verifications, it was found that the main reason for the unsatisfactory classification results was that the study area was mostly covered by clouds during the crop reproductive period, and its wide coverage and thickness caused great interference with the spectral image data quality. Therefore, this study was changed to use NDVI data from Sentinel-2 spectral remote sensing images with better quality of three views during the crop reproductive period for classification. To further improve the classification accuracy, the other three indices shown in Table 3 were also used in the study.

**Table 3.** A few indices used in this study.

| Index | Formula | Sentinel-2 Bands |
|---|---|---|
| NDVI [39–41] | $NDVI = \frac{(\rho NIR - \rho R)}{(\rho NIR + \rho R)}$ | **B5, B4** |
| IRECI [42,43] | $IRECI = \frac{(\rho_{783} - \rho R)}{(\rho NIR / \rho_{740})}$ | **B7, B4, B5, B6** |
| EVI [44,45] | $EVI = \frac{2.5 * (\rho NIR - \rho R)}{(\rho NIR + 0.6 * \rho R - 0.75 * \rho B) + 1}$ | **B8, B4, B2** |
| NDWI [46] | $NDWI = \frac{(\rho GREEN - \rho NIR)}{(\rho GREEN + \rho NIR)}$ | **B3, B5** |

The red-edge band is a sensitive band indicating the growth status of green plants. The red edge is closely related to various physicochemical parameters of vegetation and is an important indicator band describing the pigmentation status and health of plants, so a red edge is an ideal tool for remote sensing investigation of vegetation status. Zhang Ying et al. [47] identified rice based on the difference of rice in the Sentinel-2 red-edge band. Vegetation cover and leaf area index are related; the higher the vegetation cover, the larger the leaf area index, the larger the red-edge slope, and the better the corresponding vegetation growth status. Previous studies proved that the red-edge feature can be used as a spectral feature reflecting vegetation, and the study incorporated the novel inverted red-edge chlorophyll index.

Since there are natural water systems and artificial diversion channels distributed in the study area, adding the NDWI index can better extract the scattered distributed water bodies in the plot and improve the classification accuracy. At the same time, water bodies have strong absorption of short-wave infrared (SWIR), and it is sometimes necessary to use SWIR data in agricultural remote sensing monitoring to analyze and study vegetation moisture or regional drought conditions. There are large areas of rice and upland rice planted in the park that needs to be distinguished. Therefore, based on the sensitive characteristics of the short-wave infrared (SWIR) band to vegetation moisture content, the study added Sentinel-2 SWIR (B11) band as input data and added training.

So far, this study utilized the B2~B8 bands and the B11 band of Sentinel-2 data. Compared to utilizing only a single band or only NDVI (B4, B5) bands, more features were input, which had a positive impact on the study classification and improves the classification accuracy.

### 3.2. Feature Optimization Based on Sentinel-1 SAR Microwave Remote Sensing Data

In terms of spectral features, the use of NDVI data combined with the unique phenological characteristics of the crops in the park has been able to better classify the crops from other unrelated plots. In addition, this study added multiband features of different time windows in the park to improve the classification accuracy. However, it was found that the classification of water bodies and rice, rice and upland rice, and corn and trees were still highly confusing. Especially in some areas of the park, the rice and upland rice plots were adjacent to each other, and the wood and corn plots were interspersed, which reduced the spectral separability among these vegetations and caused difficulty in crop information extraction based on spectral features. Although the color display of the different crop samples differed after parameter adjustment using Sentinel-2 spectral data with multiple false color syntheses, the distinction was not very obvious. The study was conducted for all samples of the four major crops in the park. The spectral features are shown in Table 4 and Figure 4.

**Table 4.** Time series statistics of the spectral features.

| No. | Corn | Rice | Upland Rice | Soya |
|---|---|---|---|---|
| 1_0_ndvi | 0.121498 | 0.08301 | 0.080807 | 0.114307 |
| 1_1_ndvi | 0.248013 | 0.185438 | 0.258356 | 0.208938 |
| 1_2_ndvi | 0.54536 | 0.459322 | 0.387443 | 0.381946 |
| 1_0_ndwi | -0.19292 | -0.06164 | -0.11398 | -0.18346 |
| 1_1_ndwi | -0.27006 | -0.10635 | -0.23438 | -0.26069 |
| 1_2_ndwi | -0.46459 | -0.31861 | -0.32598 | -0.35796 |
| 1_0_evi | 0.081029 | 0.040263 | 0.070725 | 0.07949 |
| 1_1_evi | 0.152158 | 0.076131 | 0.142682 | 0.133734 |
| 1_2_evi | 0.397538 | 0.253147 | 0.265957 | 0.308605 |
| 1_0_ireci | 0.048656 | 0.026142 | 0.029841 | 0.040791 |
| 1_1_ireci | 0.104199 | 0.04836 | 0.107089 | 0.093804 |
| 1_2_ireci | 0.527251 | 0.255397 | 0.229979 | 0.251112 |

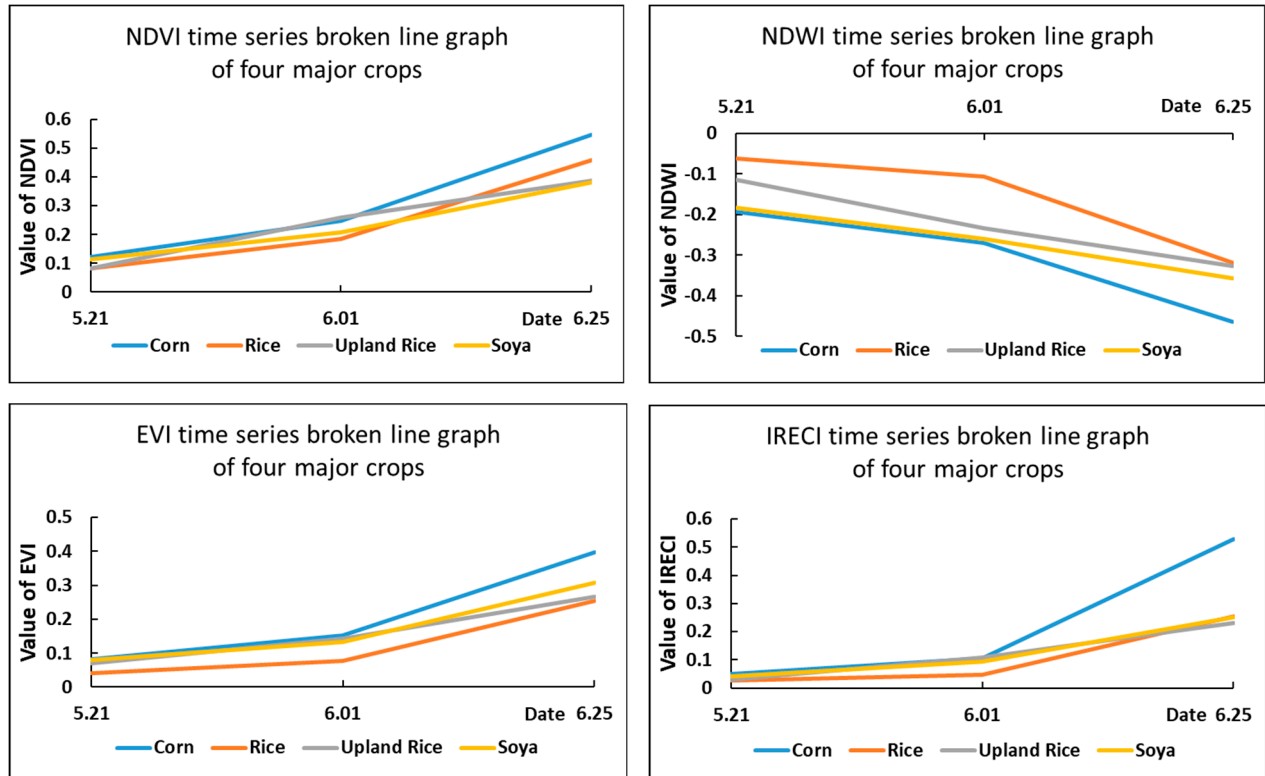

**Figure 4.** Time series broken line statistics of the spectral features.

As shown in the above graph, although the NDVI index had a good effect in distinguishing vegetation and other features, it was less obvious for distinguishing different plant (crop) types. For the NDWI index, rice and corn could be distinguished more accurately according to their water-sensitive characteristics, but the distinction between upland rice and soya was not very obvious. Similarly, the IRECI index was effective in classifying corn, but the differentiation was not obvious for the remaining three major crops of the park. Thus, although the study used as many bands as possible to classify the different crops by converting them into index features for the multiband feature of Sentinel-2, it was not able to effectively distinguish all crops, and the accuracy of the classification of the crops in the study area was not guaranteed by using spectral features alone.

Therefore, it was necessary to use other features to distinguish the different crops to improve the classification accuracy.

According to the analysis above, the classification of the crops in the park by spectral features was generally effective, and there were more misclassifications, which required the study to find other features to distinguish different crops. Based on this need, the study incorporated Sentinel-1 SAR microwave remote sensing data.

Sentinel-1 SAR data were chosen for two reasons. First, microwave remote sensing has better penetration of water vapor and clouds. To address the problem of not being able to use the time series data of the spectral features of the complete reproductive period due to the high precipitation and large cloud cover during the reproductive period of the crop in the study area, the study could be supplemented with Sentinel-1 SAR time series data to enrich the input features and, at the same time, improve the discrimination of time effects on crop growth. Second, microwave data are more sensitive to information on water bodies, soil moisture, canopy moisture, and texture. These data can be distinguished according to the morphological differences of different crops at different times.

This study used VH and VV, two polarization methods commonly used In Sentinel-1 SAR satellite images, to form VH and VV polarization images of the study area. The VH

and VV polarization characteristics of all samples using the four major crops in the park were counted to make the following statistical line graph.

As shown in Figure 5, although there are more obvious differences in the microwave characteristics of the four crops in the study area when analyzed in these three sets of images from 5 June to 17 June, in general, the microwave timing characteristics of these four major crops are roughly divided into two groups. Corn and soybean had similar curves and formed one group; rice and upland rice had similar curves and formed the second group. The differences between the two groups are obvious, but the differences between the crops within one group are smaller.

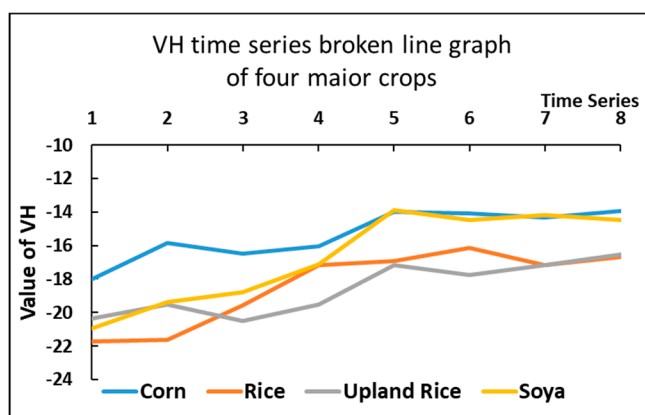 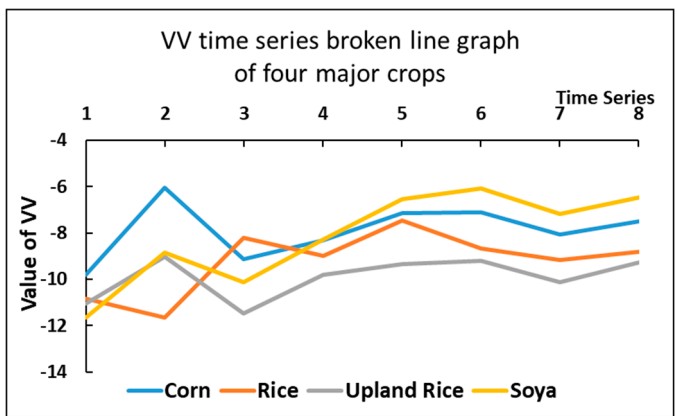

**Figure 5.** Time series broken line statistics of microwave features. Two images of similar time were averaged to form a new image. They were placed on the *x*-axis in the order of 1–8.

### 3.3. Comparison of the Pixel-Based Classification Results

As shown in the figure, the misclassification of the RF classification using Sentinel-2 images alone is more obvious. In the south and northeast of the study area, for example, the classification of rice and corn was poor in these two regions, the boundaries between them could not be clearly distinguished, and many crop plots were misclassified. As shown in Figure 6, the classification errors were significantly better when using Sentinel-2 versus Sentinel-1 SAR images for classification. However, due to the imaging characteristics of radar images, there is very serious salt-and-pepper noise and blank image element noise in the classification area of the rice plots (as shown in Figure 6(b2-detail)). At the same time, the boundaries between the different crops still did not achieve the desired effect. Although the existing image element-based denoising or filtering methods can be used to remove part of the salt-and-pepper noise, and the interpolation method can be used to supplement part of the blank image element points, the problem of indistinguishable plot boundaries still cannot be well solved.

### 3.4. Comparison of the Object-Oriented Classification Results

The SNIC method solves both of these problems well by weighting the color and spatial proximity relationships, reducing an image to a small group of interconnected superpixels while providing control over the compactness of the superpixels. This makes it possible to segment not only in color but also in grayscale. The resulting method generates compact and neat superpixels with a more realistic presentation in terms of details and more distinct boundary features.

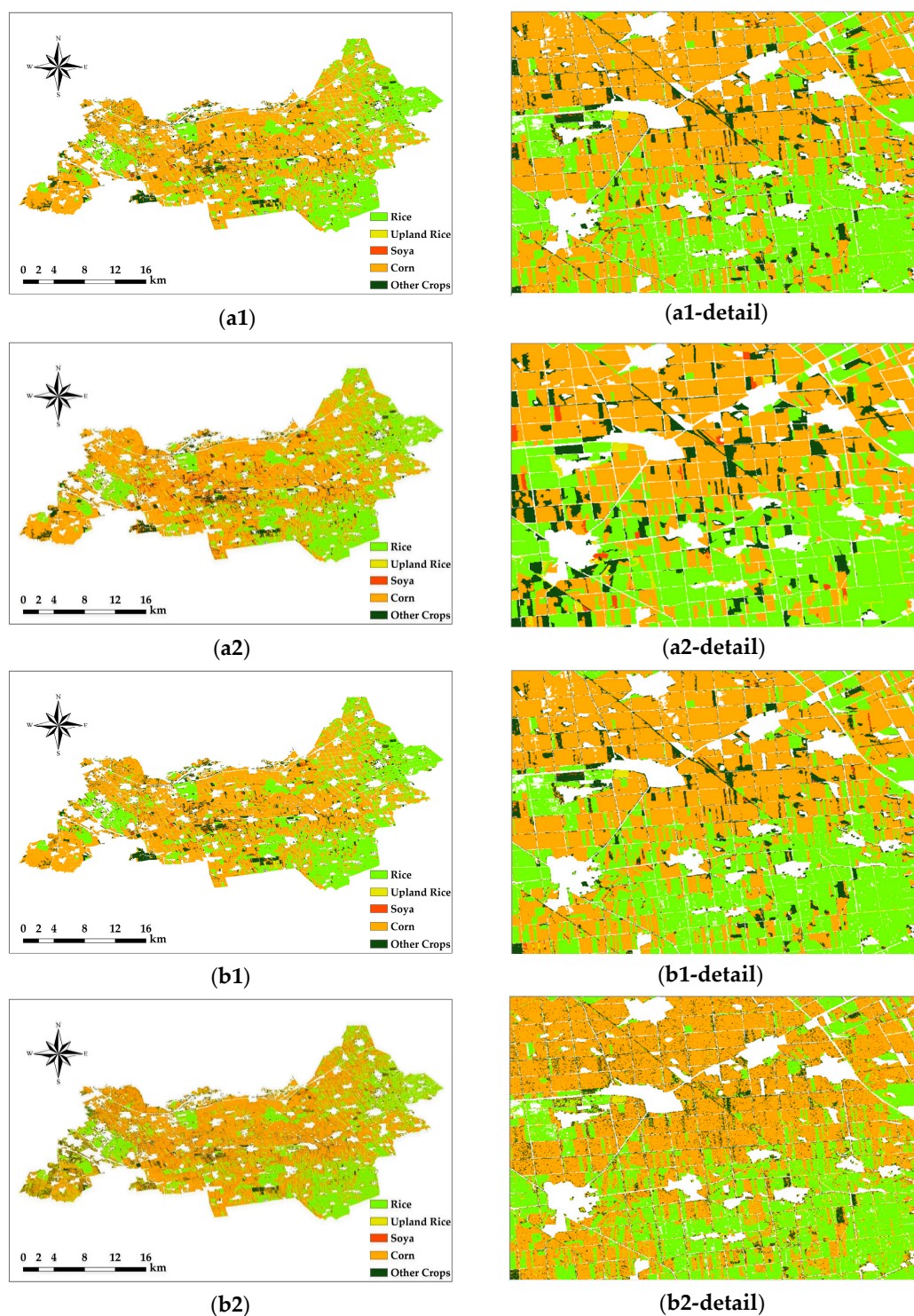

**Figure 6.** Pixel-based classification results. Only Sentinel-2 images were used in (**a1**,**a2**), Sentinel-1 SAR and Sentinel-2 images were used in (**b1**,**b2**), the RF method was used in (**a1**,**b1**), and the SVM method was used in (**a2**,**b2**). The number of each figure corresponds to that in Table 5.

In the Figure 7 comparison, the RF classification method is more sensitive for rice, and the SVM classification method is more sensitive for soya and other crops, except for the four major crops.

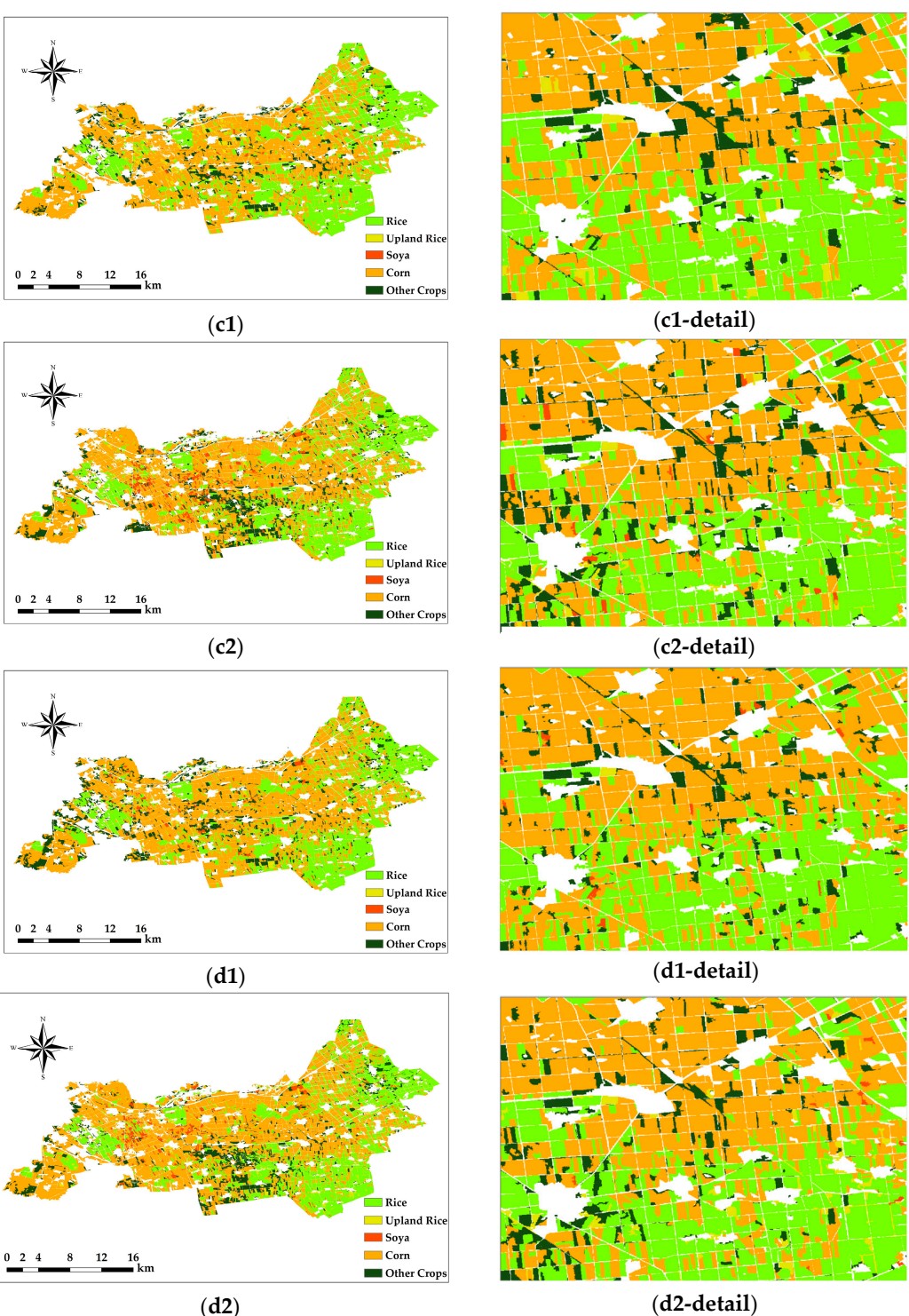

**Figure 7.** Object-oriented classification results. Pixel-based classification results. Only Sentinel-2 images were used in (**c1,c2**), Sentinel-1 SAR and Sentinel-2 images were used in (**d1,d2**), the RF method was used in (**c1,d1**), and the SVM method was used in (**c2,d2**). The number of each figure corresponds to that in Table 5.

### 3.5. Confusion Matrix and Accuracy

Table·5 and Figure 8 show the accuracy of the classification results using different methods.

**Table 5.** Summary of the accuracy.

| No. | Method | P/O | Satellite | Kappa Coefficient | Overall Accuracy | Name of Crops | Producer's Accuracy | User's Accuracy |
|---|---|---|---|---|---|---|---|---|
| a1 | RF | Pixel Based | Sentinel-2 | 0.9602 | 96.98% | Rice | 98.34% | 97.39% |
| | | | | | | Upland Rice | 95.15% | 94.90% |
| | | | | | | Soya | 89.77% | 86.72% |
| | | | | | | Corn | 98.65% | 98.39% |
| | | | | | | Other Crops | 96.06% | 98.26% |
| a2 | SVM | Pixel Based | Sentinel-2 | 0.9702 | 97.75% | Rice | 99.03% | 97.50% |
| | | | | | | Upland Rice | 98.05% | 97.78% |
| | | | | | | Soya | 91.18% | 88.83% |
| | | | | | | Corn | 99.03% | 98.82% |
| | | | | | | Other Crops | 95.87% | 98.33% |
| b1 | RF | Pixel Based | Sentinel-1 Sentinel-2 | 0.9781 | 98.35% | Rice | 99.45% | 97.37% |
| | | | | | | Upland Rice | 98.29% | 98.29% |
| | | | | | | Soya | 96.30% | 88.27% |
| | | | | | | Corn | 98.34% | 99.70% |
| | | | | | | Other Crops | 97.66% | 99.50% |
| b2 | SVM | Pixel Based | Sentinel-1 Sentinel-2 | 0.9775 | 98.29% | Rice | 98.61% | 98.84% |
| | | | | | | Upland Rice | 98.53% | 97.72% |
| | | | | | | Soya | 95.65% | 92.15% |
| | | | | | | Corn | 98.85% | 99.32% |
| | | | | | | Other Crops | 97.60% | 98.07% |
| c1 | RF | Object Oriented | Sentinel-2 | 0.9705 | 97.77% | Rice | 98.84% | 98.50% |
| | | | | | | Upland Rice | 96.05% | 94.98% |
| | | | | | | Soya | 89.68% | 89.94% |
| | | | | | | Corn | 99.31% | 99.10% |
| | | | | | | Other Crops | 97.36% | 98.50% |
| c2 | SVM | Object Oriented | Sentinel-2 | 0.9782 | 98.35% | Rice | 98.98% | 98.15% |
| | | | | | | Upland Rice | 96.89% | 97.58% |
| | | | | | | Soya | 95.86% | 96.39% |
| | | | | | | Corn | 98.73% | 98.94% |
| | | | | | | Other Crops | 98.66% | 98.61% |
| d1 | RF | Object Oriented | Sentinel-1 Sentinel-2 | 0.9823 | 98.66% | Rice | 99.52% | 98.56% |
| | | | | | | Upland Rice | 97.64% | 98.35% |
| | | | | | | Soya | 95.18% | 92.82% |
| | | | | | | Corn | 99.32% | 99.44% |
| | | | | | | Other Crops | 98.33% | 99.08% |
| d2 | SVM | Object Oriented | Sentinel-1 Sentinel-2 | 0.9803 | 98.51% | Rice | 99.31% | 98.01% |
| | | | | | | Upland Rice | 97.44% | 98.15% |
| | | | | | | Soya | 92.73% | 94.66% |
| | | | | | | Corn | 98.85% | 99.32% |
| | | | | | | Other Crops | 99.08% | 98.82% |

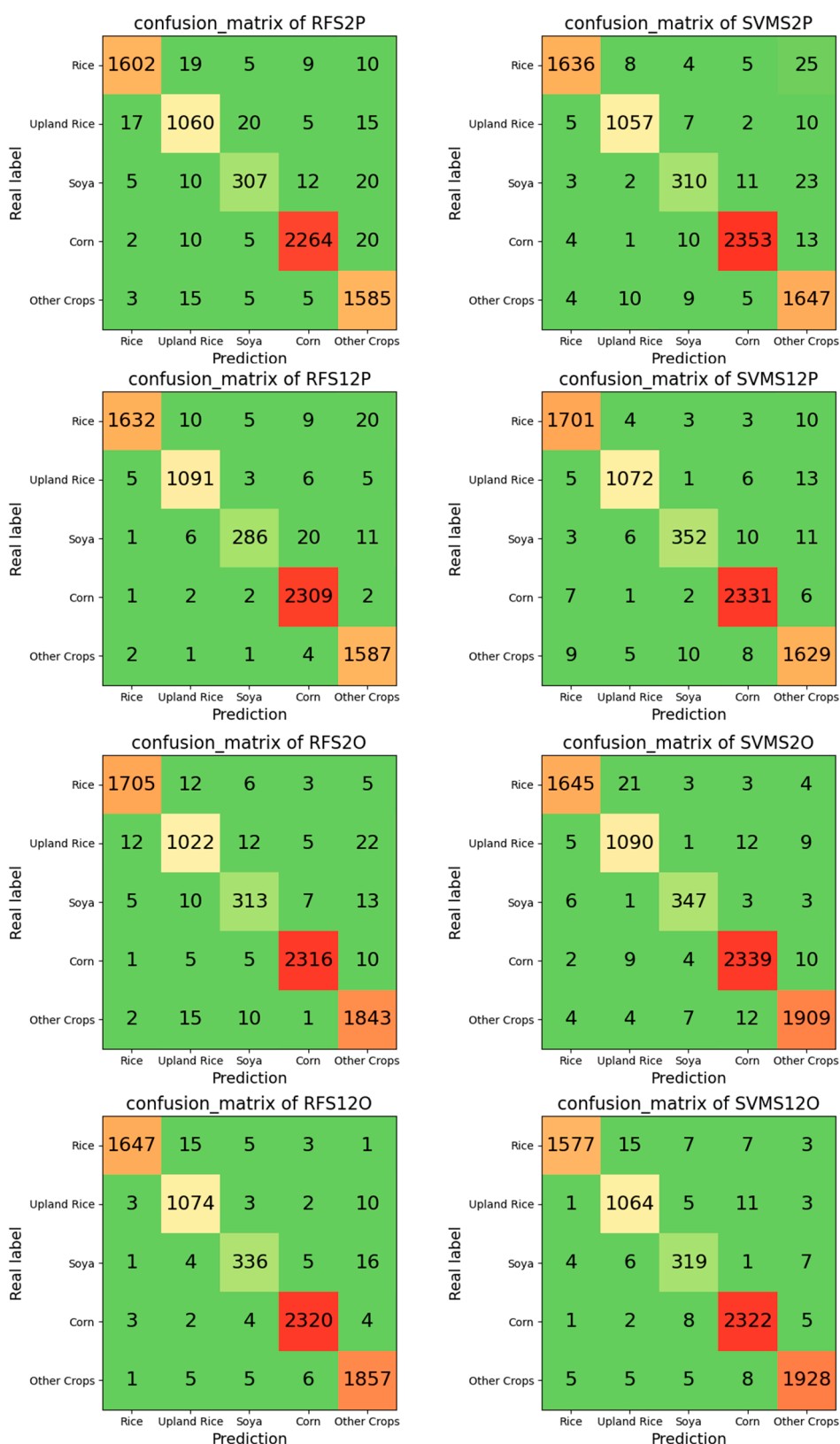

**Figure 8.** Confusion matrix of the classification results. "RF" means random forest, "SVM" means support vector machine, "S2" means only Sentinel-2 images were used, "S12" means Sentinel-1 SAR and Sentinel-2 images were used, "P" means the pixel-based method, and "O" means the object-oriented method.

## 4. Analysis and Discussion

Based on the current land use situation and the size and distribution of the crop plots in the Modern Agricultural Industrial Park in Jalaid Banner, Inner Mongolia, a suitable SNIC clustering size needs to be selected, and the SNIC images formed by using different "size" values were studied.

The "size" value used in this segmentation method needs to be flexibly selected according to the size of the study area, the spatial resolution of the remote sensor image, and the topography of the study area. After the spatial resolution of the remote sensing image is determined, the size value should be reduced appropriately for the study area with smaller plots and more dispersed crop planting distribution. For the study area with larger plots and a more concentrated spatial distribution, the size value should be increased appropriately. At the same time, the spatial resolution of the remote sensing images can be changed by image fusion and resampling, which may further enhance the segmentation effect and improve classification accuracy. When the size value is set small, the superpixels formed by the segmentation will be small, which makes misclassification occur within the parcel; when the size value is set large, the superpixels formed by the segmentation will be large, which may eventually form different parcels into one parcel for classification by mistake. Both of these cases inevitably lead to a reduction in the classification accuracy.

The following figure shows the image segmentation results under different size values. Among the three segmentation methods shown in Figure 9, when segmenting by "size = 10", the parcel segmentation is too fragmented, and the mis-segmentation is obvious in the mis-segmented part of the parcel. At the same time, segmentation by "size = 10" leads to an increase in the computational effort and computational time. In the classification results, except for soya, the accuracy of the crop classification is relatively low. This is because soya plots are relatively small and interplanted with corn. A small segmentation interval is beneficial to its classification. When segmented by "size = 30", the superpixels formed are too large, and small plots are covered and cannot be distinguished. Therefore, in this study, the segmentation was performed in the form of "size = 20". As shown in Figure 9b, when the parameter "size = 20" was used, the boundaries of the segmented superpixels overlapped more with the plot boundaries. Although, as shown in Figure 10, the classification accuracy of soya was not as good as that of "size = 10" and "size = 30", the classification accuracy of the other crops and the overall classification accuracy were better than those of the other two. Therefore, the parameter was set to "size = 20".

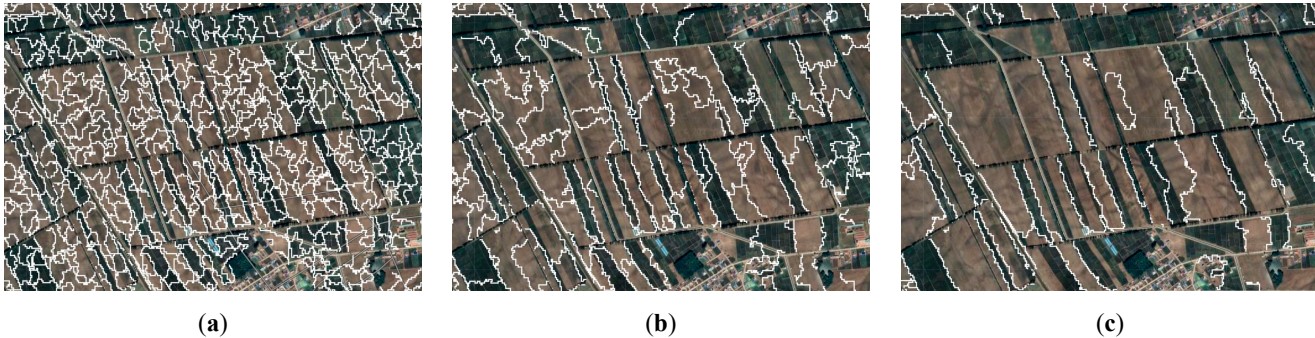

(a)　　　　　　　　　　　　　　　(b)　　　　　　　　　　　　　　　(c)

**Figure 9.** Segmentation results under different "sizes". "Size" is the superpixel seed location spacing, in pixels. (**a**) Size = 10; (**b**) Size = 20; (**c**) Size = 30.

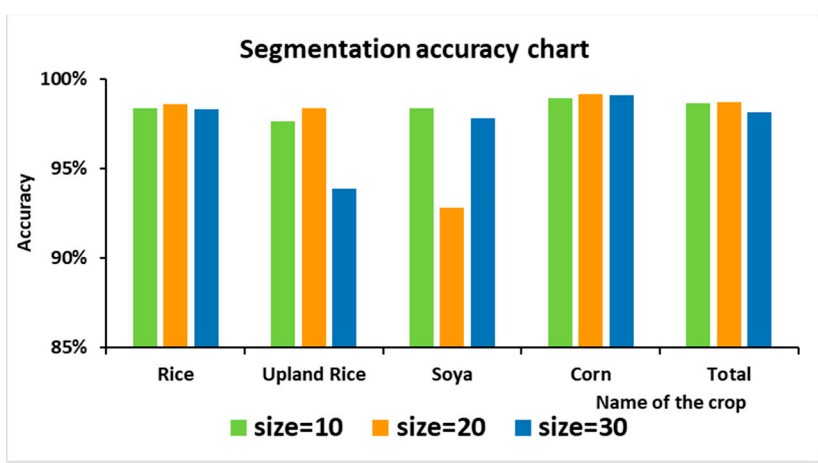

**Figure 10.** Classification accuracy statistics under different "sizes".

Combining the analysis of the results shown in Figures 6–8 and Table 5, for the image-based method, the classification accuracy was improved after adding Sentinel-1 SAR radar data with Sentinel-2 spectral data compared to using only Sentinel-2 spectral data. At the same time, the classification accuracy using RF was higher than that using SVM. The study selected areas with richer crop species and more regular plots to show the details, and the salt-and-pepper noise existed in the classification details shown in Figure 6, and, after adding Sentinel-1 SAR radar data, the salt-and-pepper noise was more obvious. After preliminary analysis, its source is the noise formed by the anomalous values in some Sentinel-1 SAR radar data after the Sentinel-2 spectral image is declouded. These noises can be processed by subsequent means, such as smoothing and filtering [48–52], but the processing time is long, while part of the boundary information may be lost.

For the object-oriented approach, the RF-based classification using Sentinel-1 SAR radar images with Sentinel-2 spectral images showed the highest accuracy of the classification results. Figure 7 shows the details of the classification results of the object-oriented approach. The salt-and-pepper noise problem was solved in the results, and the boundary information of the parcel was more complete than that of the pixel-based method. At the same time, the classification accuracy was significantly improved compared with the image-based classification method.

Due to the limitations of the data and methods, this study still had some shortcomings. Firstly, for the classification work, the number and quality of the samples have a great influence on the classification results. In this study, except for the four major crops, the sample data quantity of the remaining types of crops was small, and it was not possible to form a separate set of samples for each other crop into the trainer to ensure the sample quantity and distribution requirements. Therefore, this study used the remaining several crops as one class, which had a greater impact on the classification results. As shown in the results of Figure 6, the two methods of RF and SVM produced some differences in the classification of other crops. Secondly, the absence of Sentinel-2 spectral data led to the study's inability to extract crop spectral features using complete fertility spectral images, and the control effect of the classification results based on Sentinel-2 spectral data was reduced. Finally, the study area is flat, and the plots are relatively regular, which has good conditions for the study. However, the classification of the study under this condition could not be relocated to areas with complex topographic features. In addition, the plot boundary data used in the study greatly reduced the influence of extraneous features, and the applicability of the classification method described above in other areas lacking plot boundary data needs to be further investigated.

## 5. Conclusions

In all eight groups of results, the classification accuracy was higher after using Sentinel-1 SAR data and Sentinel-2 data than after using Sentinel-2 data only. Among the four sets of results based on the image element method, the RF-based classification method had the highest accuracy of 98.35% with a kappa coefficient of 0.9781. For rice, upland rice, corn, and other crops, the RF classification accuracy was slightly higher than the SVM classification method based on the same conditions. For soya, the SVM classification accuracy was higher than that of the RF classification method based on the same conditions, and the accuracy improvement was greater than 2%. This shows that in most cases, the classification using Sentinel-1 SAR radar data with Sentinel-2 spectral data was better than the classification using Sentinel-2 spectral data only; the RF-based classification method is better than the SVM-based classification method.

**Author Contributions:** Methodology, H.X.; software, H.X.; validation, H.X.; formal analysis, H.X.; investigation, H.X. and X.X. resources, H.X., X.X. and Q.Z.; data curation, G.Y., H.L. (Huiling Long), H.L. (Heli Li), X.Y., J.Z., Y.Y., S.X., M.Y. and Y.L.; writing—original draft preparation, H.X.; writing—review and editing, H.X. and X.X.; visualization, H.X.; supervision, X.X., Q.Z. and G.Y.; project administration, X.X. All authors have read and agreed to the published version of the manuscript.

**Funding:** This research was funded by the National Key Research and Development Program of China, grant number: 2019YFE0125300, the Special Project for Building Scientific and Technological Innovation Capacity of Beijing Academy of Agricultural and Forestry Sciences, grant number: KJCX20210433, and the National Modern Agricultural Industry Technology System, grant number: CARS-03.

**Data Availability Statement:** The data that support the findings of this study are available from the corresponding author, upon reasonable request.

**Acknowledgments:** The authors thank Google Earth Engine for providing computer resources and Sentinel-1/2 data.

**Conflicts of Interest:** The authors declare no conflict of interest.

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
