# Peer review of "Object-Oriented Crop Classification Using Time Series Sentinel Images from Google Earth Engine"

_remotesensing, doi:10.3390/rs15051353_

Round 1
Reviewer 1 Report (Previous Reviewer 2)
Thank you for incorporating the changes, I think now the paper is ready for publication.
Author Response
Response to Reviewer 1 Comments
Thank you for your approval of our study!
Hanyu Xue and co-authors
****************************************************
Correspondence information: Hanyu Xue
Key Laboratory of Quantitative Remote Sensing in Agriculture of Ministry of Agriculture and Rural Affairs, Information Technology Research Center, Beijing Academy of Agriculture and Forestry Sciences, Beijing 100097, China;
E-mail: 2222116047@stmail.ujs.edu.cn
Telephone: +8610-51503676
****************************************************

Reviewer 2 Report (New Reviewer)
Based on the time series spectral and radar features of different crop growth periods, this study combined a variety of classification methods, analyzed and discussed different methods (Methods of machine learning: RF, SVM; methods of classification: pixel-based, SNIC) in detail, and compared the classification results. It shows that SNIC method coupled with some machine learning methods not only improves the classification accuracy, but also effectively solves the problem of salt and pepper phenomenon in classification maps. The study falls well within the scope of this journal and its results and analyses are credible, I think that the study deserved to publication in this journal.
However, I have some concerns as following, and think it needs minor modifications for publication.
1. The abstract needs to be refined: Further explain the difference between SNIC and pixel-based method in classification.
2. It is suggested to combine the part of parameter setting (line450) with Section 2.2.3 /2.2.4. Make validation strategy a separate Section.
3. Add citation about NDWI. (line498).
4. Please give the basis for SNIC segmentation, that is, which kind of images were used for segmentation in the study. (line667)
5.I think all the line diagrams should be more beautiful.
Author Response
Response to Reviewer 2 Comments
Thank you for your suggestions.
We read all your suggestions carefully and revised the manuscript in response to them.
Point 1: 1. The abstract needs to be refined: Further explain the difference between SNIC and pixel-based classification methods.
Response 1:
We revised this part of the abstract.
“Compared with the pixel-based method, the combination of SNIC multi-scale segmentation and random forest classification based on time-series radar and optical remote sensing images can effectively reduce the salt and pepper phenomenon in classification, and improve crop classification accuracy with the highest accuracy of 98.66 and Kappa coefficient of 0.9823.”(line22)
Point 2: It is suggested to combine the part of parameter setting (line 450) with Section 2.2.3 /2.2.4. Make validation strategy a separate Section.
Response 2:
We moved the parameter Settings section to Senction2.2.3/2.2.4.
(line311)/(line343)
Point 3: Add citation about NDWI. (line498).
Response 3:
Crop plot boundary data was used in this study, which greatly reduced the disturbance of the water system to classification. The characteristics of rice fields allow for more precise classification. So we use NDWI. We add citation about NDWI.(line 383)
Point 4: Please give the basis for SNIC segmentation, that is, which kind of images were used for segmentation in the study.
Response 4:
We made an image collection and put all images into the SNIC algorithm, and the SNIC segmented them based on the correlation between the pixels of these images.
“The image collection of all Sentinel images was input. According to the correlation between the pixels, the images were segmented by the SNIC algorithm.” (line283)
Point 5: I think all the line diagrams should be more beautiful.
Response 5:
We reworked the diagrams. We mainly changed the font size and color. They are more beautiful now. (Figure 4,5,9)
Hanyu Xue and co-authors
****************************************************
Correspondence information: Hanyu Xue
Key Laboratory of Quantitative Remote Sensing in Agriculture of Ministry of Agriculture and Rural Affairs, Information Technology Research Center, Beijing Academy of Agriculture and Forestry Sciences, Beijing 100097, China;
E-mail: 2222116047@stmail.ujs.edu.cn
Telephone: +8610-51503676
****************************************************

Round 2
Reviewer 2 Report (New Reviewer)
The manuscript can be accepted.
This manuscript is a resubmission of an earlier submission. The following is a list of the peer review reports and author responses from that submission.
Round 1
Reviewer 1 Report
In the section of the introduction, the authors listed many previous studies, but there are several points that should have been explained according to these studies. First, why GEE should be applied in crop classification? Second, what are the advantages and disadvantages of the previous studies? Third, why different types of images should be used in the classification? Besides, the "salt-and-pepper noise" was mentioned, but does this effect exist in the classification based on images of a 30-m resolution? Please clarify.
The quality of the figures in this manuscript is quite poor and must be improved.
Figure 1 and figure 2 could be merged.
Table 1 and Figure 3 are not informative enough. In Table 1, I think if the dates of images were related to the crop phenology, the readers could be more interested in it.
Figure 4 could be improved to be with a more succinct style but with more information.
Figure 7 and Figure 8 could be merged. Besides, the meanings of a1, a2, b1 and b2 should be explained in the figure caption. The same to figure 9 and figure 10.
In figures 11 and 12, what “size” means should be clarified in the figure captions.
In the section of the study area, information on crop planting, types, and phenology should be introduced.
The parameter setting of the machine learning algorithms, i.e., random forest and support vector machine, should be clarified.
The writing and citations in this manuscript should be improved. Some examples are listed as follows.
Line 31, what does "large scale" refer to, global or regional? Please define it clearly.
Lines 41-42, please rephrase this sentence.
Lines 136-137, is there any citation?
Author Response
You can see the detailed version in my cover letter. Please see the attachment.
Response to Reviewer 1 Comments
Thank you for your suggestions, which will make my research more complete.
I read all your suggestions carefully and revised my manuscript in response to them.
Point 1: In the section of the introduction, the authors listed many previous studies, but there are several points that should have been explained according to these studies.
First, why GEE should be applied in crop classification?
Second, what are the advantages and disadvantages of the previous studies?
Third, why different types of images should be used in the classification?
Besides, the "salt-and-pepper noise" was mentioned, but does this effect exist in the classification based on images of a 30-m resolution? Please clarify.
Response 1:
- I add a new paragraph in the introduction about the reason for using GEE in my new manuscript.
“Google Earth Engine (GEE) is a tool developed by Google which stores publicly available remote sensing image data based on its millions of servers around the world and the start-of-the-art cloud-computing and storage capability, enabling GEE users to easily extract, call and analyze massive remote sensing big data resources, providing huge potential for large-scale and long-term remote sensing analysis[19][20].”
- I add a few sentences to illustrate the advantages and disadvantages.
“The above multi-temporal and multi-feature classification methods based on pixels often carry out crop classification and recognition by extracting the temporal optical (or microwave) features of image elements. They used several methods and tested them to find the best. Although they achieve high classification accuracy to a certain extent, they usually ignore the spatial correlation between adjacent image elements [7], which is prone to the salt-and-pepper noise”
- The reason for using different types of images was an important part of my study. I have illustrated it in Section 3. Image feature analysis and study results.
Firstly, the two images have complementary advantages, S1 images provide texture features and S2 images provide spectral features. Secondly, in this study, the quality of S2 spectral images during the main growth period of crops was poor, and only three images were available. In order to improve the accuracy of classification, S1 images were added. So different types of images were used in this study.
- Salt-and-pepper noise, in my experience, exists in most pixel-based classifications with high resolution images(Sentinel-1,Sentinel-2 and Landsat-8 have SAP noise.). Its essence is the misclassification of pixels affected by various reasons. In my study, when I was experimenting with pixel-based classification, Salt-and-pepper noise phenomenon existed and was widely distributed in the study area. So I used SNIC to eliminate this phenomenon as much as possible. But this phenomenon may not be obvious for lower resolution images.
Point 2:
The quality of the figures in this manuscript is quite poor and must be improved
Figure 1 and figure 2 could be merged.
Figure 4 could be improved to be with a more succinct style but with more information.
Figure 7 and Figure 8 could be merged. Besides, the meanings of a1, a2, b1 and b2 should be explained in the figure caption. The same to figure 9 and figure 10.
In figures 11 and 12, what “size” means should be clarified in the figure captions.
Response 2: Thank you for your suggestions. These suggestions are precious. I rechecked my figures and I modified the figures according to your suggestions. I merged all figures you mentioned. And I add explanations for them.
Point 3: Table 1 and Figure 3 are not informative enough. In Table 1, I think if the dates of images were related to the crop phenology, the readers could be more interested in it.
Response 3: The point you mentioned is very important. I add Table 1. to show the phenology of the main corps.
Point 4: The parameter setting of the machine learning algorithms, i.e., random forest and support vector machine, should be clarified.
Response 4:I perfected the classification methods and classification results. The confusion matrix and accuracy are added. They are shown in Figure 8 and Table 5.
Point 5: The writing and citations in this manuscript should be improved. Some examples are listed as follows.
Line 31, what does "large scale" refer to, global or regional? Please define it clearly.
Lines 41-42, please rephrase this sentence.
Response 5:
(1) The greater the range of access to information, the better the decision-making. For this study, the ‘large-scale’ here does have some ambiguity. I changed it to "regional."
(2) I rewrote the sentence into “Currently, crop classification and identification methods using time series images using image-oriented are widely applied [1-5]. HJ-CCD time-series optical remote sensing images were used to construct a decision tree based classification model with the spectral and vegetation index time-series variation characteristics of crops for effective classification and identification of multiple crop plantings. [1] Multi-temporal RADARSAT-2 fully polarized SAR time series images were used to achieve efficient extraction of the phenological period of rice based on the time-series curve variation characteristics of polarization feature parameters of rice.”
Point 6: Lines 136-137, is there any citation?
Response 6: The strong penetration of SAR is due to its microwave properties. Microwaves penetrate the cloud and atmosphere very well. I provide a citation for this.

Reviewer 2 Report
Overall, an interesting study and the results are presented meaningfully.
However, major changes must be made as suggested:
- Description of validation strategy is missing completely and thus, the classification accuracy results are not adequatley presented yet
- Moderate English changes required
- Better depiction and description of the utilized feature datasets
Abstract, rephrase: The images of classification result --> The resulting maps of land use classification...
line 122: rephrase declouded (e.g. masked out)
line 125: rephrase to "swath"
section 2.2.1: was the Sentinel-1 data pre-processed (GRD backscattering, terrain correction and radiometric calibration), please add information on that
line 182: rephrase to "Then the crop types in the study area are classified..."
line 192: rephrase to backscattering or roughness coefficient
line 218: remove Deepl reference
section 2.2: please state how the ground truth data was split and used for training and for validation.
Please also include a figure with the distribution of training/validation samples per crop type
Figure 5: if data is available, please include more dates (data points) of the respective spectral index time series, 3 dates is not enough
Figure 6: include the date labels in the x-axis
Figures 7-9: include a brief description and subtitles of the subplots (a1,a2,b1,b2,c1,c2 and so on)
section 4, Table 4: include a classification accuracy confusion matrix to better depict inter-class misclassifications with recall, precision (user/producer accuracy)
line 321: "impossible" - why? please provide proof e.g by comparing classification accuracies between different set of features NDVI vs. spectral/SAR features
please add 4.2 "Conclusion" or put it another section 5
Author Response
You can see the detailed version in my cover letter. Please see the attachment.
Response to Reviewer 2 Comments
Thank you for your suggestions on my work! These suggestions are very meaningful to me.
I have read all your suggestions carefully。
I will revise my manuscript according to your suggestions.
Point 1: Abstract, rephrase: The images of classification result --> The resulting maps of land use classification...
Response 1: I have revised my account.
Point 2: line 122: rephrase declouded (e.g. masked out).
Response 2: I rephrase it to “the part covered by cloud must be removed.”
Point 3: line 125: rephrase to "swath"
Response 3: I did get the proper term wrong here. I rephrase it to” (Stripmap(SM), Interferometric Wide Swath(IW), Extra Wide Swath(EW) and Wave(WV))”. I will pay attention to the use of WORD.
Point 4: section 2.2.1: was the Sentinel-1 data pre-processed (GRD backscattering, terrain correction and radiometric calibration), please add information on that.
Response 4:I used Sentinel-1 SAR data and I pre-processed all images with terrain correction(SRTM DEM 30m data). I have show this in Paragraph 4 ,Section 2.1.2.
Sentinel-1 images I used are S1_GRD. I revised Table2.
Point 5: line 182: rephrase to "Then the crop types in the study area are classified..."
Response 5: “The advantage of this approach is that it eliminates a variety of irrelevant land use types in the study area, reduces the complexity of classification due to the input of other irrelevant features such as residential houses, roads, trees and water systems, and then the crop types in the study area are classified with a higher accuracy.”
Point 6: line 192: rephrase to backscattering or roughness coefficient
Response 6:It is “backscattering”
Point 7: remove Deepl reference
Response 7: remove Deepl reference
Point 8: (1)section 2.2: please state how the ground truth data was split and used for training and for validation. (2)section 4, Table 4: include a classification accuracy confusion matrix to better depict inter-class misclassifications with recall, precision (user/producer accuracy)
Please also include a figure with the distribution of training/validation samples per crop type.
Response 8: These samples were randomly selected in the proportion of 7:3. ”7” of training data and “3” of test data. Every crop type has its accuracy confusion matrix. In my submission, the manuscript will be too long if these matrices are displayed in it.
There are eight groups of tables like this.
I added “user’s Accuray” and “producer’s Accuracy” two new columns to the table. This can also show accuracy information clearly.
Point 9: Figure 5: if data is available, please include more dates (data points) of the respective spectral index time series, 3 dates is not enough.
Response 9: The reason why I used only 3 Sentinel-2 images is the local weather conditions there is bad. I checked all images from 01 June to 30 September and most of these images are coverd by cloud. Their coverage is too high to use. I have briefly explained this problem here Paragraph 1 ,Section 2.1.2.
Point 10: Figure 6: include the date labels in the x-axis.
Response 10:
I really didn't make it clear in my manuscript. I used 16 Sentinel-1 SAR images(table2) in my study. In order to avoid the impact of noise as much as possible. I took the average of the two images. That was why I only showed 8 dates in Figure 5.
I will revise it and give it a clear explanation in my manuscript.
“Figure 5. Time series broken line statistics of microwave features. Two images of similar time are averaged to form a new image. They are numbered 1-8 in sequence.”
Point 11: Figures 7-9: include a brief description and subtitles of the subplots (a1,a2,b1,b2,c1,c2 and so on)
Response 11: Thank you for your suggestion. This suggestion is very valuable. I rechecked my figures and I modified the figures according to your suggestions. And I add explanations for them.
